# Discrimination and violence against women migrant workers in Thailand during the COVID-19 pandemic: A mixed-methods study

Montakarn Chuemchit[1]ᴼ*, Nyan Linn[2]ᴼ, Chit Pyae Pyae Han[2], Zayar Lynn[2], Suttharuethai Chernkwanma[2], Nutta Taneepanichskul[2], Wandee Sirichokchatchawan[2], Ratana Somrongthong[2]

1 College of Public Health Sciences, Chulalongkorn University and Excellent Center for Health and Social Sciences and Addiction Research, Bangkok, Thailand, 2 College of Public Health Sciences Chulalongkorn University, Bangkok, Thailand

ᴼ These authors contributed equally to this work.
* Montakarn.Ch@chula.ac.th

**Data Availability Statement:** All relevant data are within the paper and its Supporting Information files. Quantitative data are openly accessible.

## Abstract

### Background

Women migrant workers are vulnerable to discrimination and violence, which are significant public health problems. These situations may have been intensified during the COVID-19 pandemic. This study aimed to investigate discrimination against women migrant workers in Thailand during the COVID-19 pandemic and its intersection with their experiences of violence and associated factors.

### Methods

A mixed-methods study design was employed to collect data from 572 women migrant workers from Myanmar, Lao People's Democratic Republic, and Cambodia. Face-to-face interviews were conducted with 494 participants using a structured questionnaire for quantitative data, whereas qualitative data was collected through 24 in-depth interviews and focus group discussions with 54 migrant women. Simple and multiple logistic regression and content analysis were employed.

### Results

This study found that about one in five women migrant workers experienced discrimination during the COVID-19 pandemic. Among those who experienced discrimination, 63.2% had experienced intimate partner violence and 76.4% had experienced non-intimate partner violence in their lifetime. The multivariable analysis revealed that women migrant workers who had experienced any violence (AOR = 2.76, 95% CI = 1.49, 5.12), lost their jobs or income during the pandemic (AOR = 3.99, 95% CI = 2.09, 7.62), and were from Myanmar (AOR = 4.68, 95% CI = 1.79, 12.21) were more likely to have experienced discrimination.

However, access to qualitative data, specifically interview transcripts, is restricted due to the sensitive nature of the content relating to experiences of violence. In our "Participant Information Sheet and Consent Form," we emphasized confidentiality, stating: "...Information directly related to participants will remain confidential. Only the research team will have access to participants' information. The results of the study will be reported as a total picture. Any information which could be able to identify you will not appear in the report." Access to qualitative data may be granted in a de-identified format or through excerpts of interview responses. This study received ethical approval from the Research Ethics Review Committee for Research Involving Human Research Participants, Group I, Chulalongkorn University, Thailand (COA No. 232/2020). For inquiries regarding access to field data and long-term data accessibility, please reach out to the Regional Programme Manager at UN Women Regional Office for Asia and the Pacific at Email: melissa.alvarado@unwomen.org, Website: //asiapacific.unwomen.org. The dataset in question is titled " Qualitative data of violence among WMW in Thailand.

**Funding:** This study was supported by the EU funded Spotlight initiative, Safe and Fair Program, UN Women Thailand (MC; PSA-ROAP-2021-008). https://asiapacific.unwomen.org/en/focus-areas/end-violence-against-women/safe-and-fair The funders had no role in data collection and analysis, decision to publish, or preparation of the manuscript.

**Competing interests:** The authors have declared that no competing interests exist.

## Conclusion

The results suggest that the intersection of discrimination and violence against women migrant workers in Thailand demands special interest to understand and address the problem. It is recommended that policymakers provide interventions and programs that are inclusive and responsive to the unique needs of women migrants depending on their country of origin and job profile.

## Introduction

Women migrant workers are vulnerable to discrimination and violence based on their "race, ethnicity, nationality, age, migration status or other sex- or gender-associated characteristics" [1]. Although they contribute significantly to the economy of Thailand, they often face low pay, long hours, and little social safety from both employers and society [2]. The COVID-19 pandemic and lockdown measures in Thailand exacerbated this situation; many women migrant workers lost their work and legal status because of economic downturns. During the pandemic, they often experienced discrimination and violence related to their nationality and perceived role in spreading the virus [3–5]. However, comprehensive research on discrimination against women migrant workers is still inadequate, especially among women migrant workers [6,7].

Discrimination and violence against women migrant workers are major public health problems globally [7]. The negative consequences include poor mental health, reproductive health problems, chronic pain, post-traumatic stress disorder (PTSD), and an increased risk of contracting sexually transmitted infections (STIs) [8]. These issues can also lead to social isolation and reduced access to healthcare and legal services [9,10], and they are serious human rights violations that hinder the achievement of the UN's Sustainable Development Goals (SDGs), particularly Goal 5, gender equality and empowerment of all women and girls, and Goal 8, promotion of inclusive and sustainable economic growth, full and productive employment, and decent work for all [11]. In Thailand, the basic human rights of women migrant workers are often neglected, with incidences of discrimination and violence occurring against them [3–5].

This study hypothesizes that women migrant workers who have had experiences of violence are more likely to face discrimination in the context of COVID-19 pandemic. Intersectionality theory uncovers the sociological analytical framework that explains how discrimination intersects with violence and other social identities, such as nationality and job profile, shaping the experiences of women migrant workers. Therefore, women who have experienced violence may face discrimination because of their intersecting identities [12]. Social learning theory suggests that people learn behavior from their environment; as a result of past experiences of violence, women may learn to expect negative treatment and discrimination from others [13]. The stereotype content model proposes that people categorize others based on two dimensions: warmth (friendliness, kindness) and competence (intelligence, skill); this means that women who have experienced violence may be seen as less competent and less deserving of respect and fair treatment [14].

Regarding legal and policy frameworks for migrant workers, the Thai government has made efforts to prohibit discrimination based on gender, nationality, race, and religion in employment, such as the 2017 Royal Ordinance on Management of the Employment of Aliens [15]. However, reports indicate that women migrant workers in Thailand still face several barriers to accessing justice and that their access to equal protection and benefits depends on

several factors, such as their migration status or working sector [16–18]. The COVID-19 pandemic worsened this situation; lack of legal status, language challenges, and fear of penalties make it difficult for migrant women to access support systems and services, such as medical treatment and legal matters, and they are reluctant to report abuse. In the context of the COVID-19 crisis in Thailand, it is important to clarify that there were no emergency regulatory measures in place that would result in legal action against women migrant workers with irregular migration status seeking screening, testing, or treatment measures to mitigate the spread of the virus. While they were encouraged to approach for assistance without fear of immediate legal consequences related to irregular migration, it is essential to note that the issue of irregular migration status remains a separate matter [3–5].

Given these challenges, it is important to investigate the details of the experiences of women migrant workers in Thailand during the COVID-19 situation. While there is a growing body of literature on this issue, there is still a gap in knowledge of the intersection of discrimination and violence against these women during the pandemic. In particular, there is a lack of mixed-methods studies that could provide a more comprehensive understanding of this issue through a multidimensional and holistic approach [19,20]. Therefore, this paper aims to investigate 1) the extent of discrimination against women migrant workers in Thailand during the pandemic, 2) the intersection of this discrimination with these women's experiences of violence, and 3) the factors associated with discrimination. This study is expected to fill this gap by providing empirical evidence, which can inform policy and programs to address these problems and contribute to the development of effective interventions to protect and promote the rights and well-being of women migrant workers in Thailand.

## Materials and methods

### Study area and population

Mixed-methods research was carried out in four regions of Thailand, including seven provinces: Bangkok, Chiang Rai, Chonburi, Samut Prakan, Rayong, Samut Sakhon and Tak, where migrant workers commonly reside and work. Taking into consideration a sample size proportionate to the size of the migrant population according to their countries of origin, a snowball sampling technique was applied to reach the study population (a hard-to-reach group), which may have helped build rapport and trust with the participants. The study recruited women migrant workers who fulfilled the following eligibility criteria: being a woman migrant worker aged 18 years or above; having been a registered or non-registered resident of Thailand for at least one month; having migrated from Lao PDR, Myanmar, or Cambodia; living in Central, Northern, Western, and/or the Eastern region; currently working or seeking work in Thailand; and able to communicate in their own language. Table 1 provides details about the participants in this study.

### Data collection by enumerators

In each province, quantitative data was collected by enumerators: female migrant workers who were acquainted with and had a good understanding of the study population, had worked in

**Table 1. Participants included in the study (total = 572 women migrant workers).**

|  | Cambodia | Lao PDR | Myanmar | Total |
|---|---|---|---|---|
| **Quantitative survey (participants)** | 132 | 160 | 202 | 494 |
| **In-depth interview (participants)** | 6 | 6 | 12 | 24 |
| **Focus group discussion (participants) (6 participants in each group)** | 12 | 12 | 30 | 54 |

the area of migrant workers, and had experience in community-based research. There were 5–8 enumerators per province depending on the number of respondents. The team conducted in-person and virtual 2-day workshop training in collaboration with partner organizations. Before data collection began, the women migrant workers who served as enumerators received comprehensive training, which included background information, the objectives of the study and its duration, eligibility criteria for respondents, how to approach the respondents, gender-responsive interviewing, ethical concerns, the survey questionnaire, and how to respond to violence. Women migrant workers were the best candidates to carry out data collection because they are familiar with and sensitive to the difficulties of women in their own communities. This approach eliminated the language barrier, and enumerators were empowered with paid employment, skill development, and research experiences. The enumerators would gain an improved understanding of gender equality, gender-based violence, and other gender-related ideas as a result of the training. In addition, this method fostered a strong sense of ownership of the research and its outcomes. Moreover, during data collection, the researchers and enumerators held wrap-up meetings to discuss the day's data collection activities. These meetings provided an opportunity to review and discuss the data collection process, address any issues that arose, and ensure consistency and accuracy in the overall data collection effort.

The participants in each province were approached in collaboration with partner CSOs/NGOs, and eligible participants were invited to the interview location. Each participant was interviewed face-to-face by a well-trained enumerator in a private place to ensure their confidentiality and safety, which reduced hesitation to disclose their information. At any time during the interview, the respondents were able to pause, skip the question, or stop answering. An information sheet consisting of health care services, call center numbers, social support and counseling services, police stations, and the contact information of embassies was provided to each participant after the interview. Upon request, additional or urgent support from local health authorities was provided to participants by the enumerators.

Focus group discussions (FGDs) and in-depth interviews were conducted by the research team, which included experienced researchers with expertise in qualitative data collection. This ensured that the data collected was of high quality and that participants could provide rich and meaningful insights.

## Research instruments

The quantitative questionnaire was developed in the languages of English, Khmer, Lao, and Myanmar and comprised seven sections: 1) Participant's Information, 2) Gender Attitudes, adapted from Gender Equitable Men Scale [21], 3) Impact of COVID-19 Pandemic, 4) General Safety, adapted from the WHO multi-country study on women's health and domestic violence [22], which is both standardized and freely accessible for use, 5) Impact of Violence, 6) Coping and Knowledge, and 7) Socio-Economic Situation.

Regarding discrimination, section 3 included the question, "Did anyone treat you badly or discriminate against you during the Coronavirus Outbreak (March 2020 until now)?" Respondents who answered "Yes" were asked three further questions: "Do you think that they discriminated against you or treated you badly because you were a migrant worker?", "Do you think they have treated you badly or discriminated against you because of your age?", and "Do you think that because you are a woman, they treated you badly or discriminated against you?" The response to each question was "Yes" or "No." The pre-test was done, and the Kuder-Richardson score was 0.8.

In section 4, the women's experiences of violence were asked in two parts: intimate partner violence and non-intimate partner violence. The following forms of violence were asked about

in each part: 1) economic (restricting working or making money, forcibly taking income, and expelling from residence), 2) cyber bullying (insulting, harassing, and threatening online), 3) psychological (insulting or humiliating, scaring, and threatening); 4) physical (slapping, throwing, punching, beating, dragging, choking, burning, and threatening to use a weapon), and 5) sexual (threatening or coercing to have sex against her will, having unwanted sex because of fear, and being forced to have sex that was disgusting or shameful for her). Respondents were asked about the frequency at which they had experienced each act of violence (once, a few times, or many times) and about the timing (whether it occurred within the past 12 months). The pre-test was done, and the Cronbach's alpha scores were 0.8 and 0.9, respectively.

In-depth interviews using a semi-structured questionnaire and FGDs were conducted to collect qualitative data. They consisted of questions about experiences of violence, the context of this violence and its intersection with discrimination, and the impact of COVID-19 on participants' lives and experiences of violence, particularly in the context of labor migration. We conducted pilot testing to evaluate and refine the qualitative research instruments and methodologies prior to the full-scale investigation. During this preliminary stage, a small group of representative participants was recruited to participate in interviews, allowing for an assessment of the instruments' clarity, cultural sensitivity, and effectiveness.

## Ethical considerations

The data collection was conducted from 2021–2022 after being approved by the Ethics Review Committee for Research Involving Human Research Subjects, Health Sciences Group, Chulalongkorn University (COA.No.232/2020), Thailand. Informed consent was obtained from all participants, written or verbally for those who did not want to sign their name on a written consent form. Participants were kept anonymous, and their involvement was completely voluntary. Confidentiality of information was preserved throughout the study by using codes instead of the participants' names. Participants had the option to withdraw from the study at any point during the recruitment and interview process. Furthermore, those individuals who were recruited but did not complete the interview received the same treatment offered to respondents who successfully participated in the study.

## Data analysis

The data entry and analyses of the quantitative data were done using the Statistical Package for the Social Sciences (SPSS) software version 28.0 (IBM SPSS, version 28, Armonk, NY, USA; Chulalongkorn University license). Median and interquartile range (IQR) were used to describe the continuous variables, and number and percentage (%) were used to describe the categorical variables. Inferential statistics was analyzed in two models for the following outcome: Discrimination (0 = No, 1 = Yes). Model 1 used bivariate analysis (simple logistic regression) to identify the independent association between each variable and discrimination; the results are presented as Crude odds ratios (COR) and 95% confidence intervals (CI). Variables with a p-value < 0.2 from bivariate analysis were potential predictors and included in Model 2, which used multivariable analysis (multiple logistic regression) to investigate the factors associated with discrimination while adjusting other variables [23]; the results were presented as adjusted odds ratios (AOR) and had a 95% CI. The threshold for significance (alpha) in this study was 0.05.

For the qualitative data, the researchers transcribed the in-depth interviews and FGDs verbatim by listening to the tape recordings to verify the accuracy of the data. The data were categorized under main themes (e.g., migration journey, challenges faced by women migrant

workers, support, and protection), and content analysis/thematic analysis was applied for analysis and interpretation.

## Results

### Sociodemographic characteristics and related information

The average age of the women in this study was 27 ± 13 years, and most of them were in the range of 18–30 years (38.7%). About one in 10 of the women had not received any education (12.1%), while only one in 20 had received a university level of education (5.5%). More than half of the women were married (68.2%). Nearly half of the women came from Myanmar (40.9%), and four in 10 women had irregular migration status (40.3%). Additionally, about 1 in 5 of the women received 5,000 baht or less as their average monthly income (18.2%). During the COVID-19 pandemic, more than half of the women in this study lost their job or income (58.3%). The details are presented in Table 2. The qualitative data confirms that during the pandemic, most women migrant workers were affected by lockdowns, quarantine, or both. More than half of those affected by lockdowns or quarantine felt insecure and had concerns about finances, stress or anxiety about the future, uncertainty about the virus, worries about their living space, and concerns about their visa status.

### The extent of discrimination against women migrant workers during the COVID-19 pandemic

The overall percentage of women migrant workers who had experienced discrimination was found to be 18.0%. Among the three forms of discrimination investigated in this study, discrimination because of the respondent's status as a migrant worker was the most common (94.4% of women reported being discriminated against; Table 3). In addition, from the qualitative data, we found that women migrant workers who returned to their home countries during the pandemic often faced hardship and sometimes discrimination. As a consequence, some women migrant workers returned to Thailand using irregular channels, which increased their vulnerability to trafficking and violence. COVID-19 generated an increase in unemployment, mental health problems, and isolation, as well as discrimination against women migrant workers. Debt bondage, economic dependency, and stress disempowered women migrant workers. These findings are consistent with the qualitative results of the study regarding the respondent's experiences of discrimination, encompassing aspects related to their migrant worker status, gender, and age. The narratives provided by participants reveal multifaceted encounters with discrimination that shed light on the intersectionality of these factors. For instance, several respondents expressed facing discrimination not only due to their migrant status but also as a consequence of being women, highlighting the compounding effects of gender-based discrimination.

### The experiences of current violence of women migrant workers during the COVID-19 pandemic

The experiences of women migrant workers regarding current violence in the past 12 months during the COVID-19 pandemic are described in Table 4. Psychological violence was the most common type (11.4% for IPV and 14.4% for N-IPV), followed by economic violence (7.7% for IPV and 7.3% for N-IPV) and sexual violence (6.9% for IPV and 3.4% for N-IPV). The proportion of cyberbullying was 6.4% for IPV and 2.2% for N-IPV, and that of physical violence was 5.9% for IPV and 0.8% for N-IPV, respectively.

**Table 2. Selected sociodemographic characteristics of women migrant workers in Thailand (n = 494).**

| Characteristics | Number | (%) |
|---|---|---|
| **Age** | | |
| •18–30 | 191 | 38.7 |
| •31–40 | 182 | 36.8 |
| •42–50 | 98 | 19.8 |
| •50–63 | 23 | 4.7 |
| •Median ± IQR | 27.00 ± 13.00 | |
| •Range | 18–63 years | |
| **Education level** | | |
| •No education | 60 | 12.1 |
| •Primary school | 176 | 35.2 |
| •Secondary school | 233 | 47.2 |
| •University and above | 27 | 5.5 |
| **Marital status** | | |
| •Single | 89 | 18.0 |
| •Married | 337 | 68.2 |
| •Divorced | 68 | 13.8 |
| **Country of origin** | | |
| •Lao PDR | 160 | 32.4 |
| •Cambodia | 132 | 26.7 |
| •Myanmar | 202 | 40.9 |
| **Migration status** | | |
| •Regular | 295 | 59.7 |
| •Irregular | 199 | 40.3 |
| **Average income (baht per month)** | | |
| •Less than or equal to 5,000 | 90 | 18.2 |
| •5,001–10,000 | 164 | 33.2 |
| •10,001–15,000 | 193 | 39.1 |
| •15,001–30,000 | 47 | 9.5 |
| **Job or income loss during the COVID-19 pandemic** | | |
| •No | 206 | 41.7 |
| •Yes | 288 | 58.3 |
| **Ability to read in Thai** | | |
| •Cannot read at all | 237 | 48.0 |
| •Can read a little bit | 183 | 37.0 |
| •Can read fluently | 74 | 15.0 |
| **Ability to speak in Thai** | | |
| •Cannot speak at all | 38 | 7.7 |
| •Can speak a little bit | 190 | 38.5 |
| •Can speak fluently | 266 | 53.8 |

## The intersection of discrimination and violence

Regarding IPV, the findings revealed that 63.2% of the women migrant workers who reported being discriminated also experienced one or more forms of IPV in their lifetime. In addition, regarding N-IPV, 76.4% of women who experienced discrimination were also survivors of N-IPV (Tables 5 and 6).

One woman migrant shared her experiences during an in-depth interview: "*I have experienced discrimination from my employer and co-workers due to my nationality and accent. The*

**Table 3. The proportion of women migrant workers in Thailand who experienced discrimination.**

| Discrimination | I do not think so | | I think so | |
|---|---|---|---|---|
| | Number | (%) | Number | (%) |
| **Being discriminated during the COVID-19 pandemic (n = 494)** | 405 | 82.0 | 89 | 18.0 |
| •**Being discriminated against because of respondent's status as a migrant worker (n = 89)** | 5 | 5.6 | 84 | 94.4 |
| •**Being discriminated against because of respondent's status as a woman (n = 89)** | 73 | 82.0 | 16 | 18.0 |
| •**Being discriminated against because of respondent's age (n = 89)** | 81 | 91.0 | 8 | 9.0 |

*employer would often give me more difficult and physically demanding tasks than other colleagues, and co-workers would mock my accent and make fun when I spoke.*" She also disclosed that she had experienced intimate partner violence in the past. She shared that her partner would often physically and emotionally abuse her and that she felt unable to leave the relationship due to financial dependence and fear of deportation (In-depth interview: I-23).

## Factors associated with discrimination against women migrant workers

As shown in Table 7, bivariate analysis revealed the statistically significant factors associated with discrimination against women migrant workers during the COVID-19 pandemic, as follows.

Women who reported having experienced violence in their lifetime showed higher odds of reporting experiences of discrimination compared to those who did not report being victims of violence (OR = 4.47, 95% CI = 2.55, 7.85).

Women aged 41–63 years reported higher odds of reporting being discriminated against than those aged 18–30 years (OR = 2.72, 95% CI = 1.52, 4.87). Women with a secondary or higher education level also had higher odds of reporting being discriminated against compared

**Table 4. The proportion of women migrant workers who experienced current violence.**

| Current Violence | % of Intimate Partner Violence (n = 405) | % of Non-Intimate Partner Violence (n = 494) |
|---|---|---|
| **Economic** | 7.7 | 7.3 |
| **Cyberbullying** | 6.4 | 2.2 |
| **Psychological** | 11.4 | 14.4 |
| **Physical** | 5.9 | 0.8 |
| **Sexual** | 6.9 | 3.4 |

**Table 5. Intersection of discrimination and experiences of intimate partner violence (n = 405).**

| Discrimination | Number (%) of Intimate Partner Violence (n = 405) | | |
|---|---|---|---|
| | No | Yes | Total |
| **No** | 205 (62.3) | 124 (37.7) | 329 (100) |
| **Yes** | 28 (36.8) | 48 (63.2) | 76 (100) |

**Table 6. Intersection of discrimination and experiences of non-intimate partner violence (n = 494).**

| Discrimination | Number (%) of Non-Intimate Partner Violence (n = 494) | | |
|---|---|---|---|
| | No | Yes | Total |
| **No** | 245 (60.5) | 160 (39.5) | 405 (100) |
| **Yes** | 21 (23.6) | 68 (76.4) | 89 (100) |

**Table 7. Factors associated with discrimination against women migrant workers (n = 494).**

| Characteristics of women migrants | Discrimination | | Model 1 | | Model 2 | |
|---|---|---|---|---|---|---|
| | No | Yes | COR (95% CI) | p–value [a] | AOR (95% CI) | p–value [a] |
| | Number (%) | Number (%) | | | | |
| **Experience of Violence** | | | | | | |
| •No | 208 (92.4) | 17 (7.6) | 1 | | 1 | |
| •Yes | 197 (73.2) | 72 (26.8) | 4.47 (2.55,7.85) | **<0.001** | 2.76 (1.49, 5.12) | **0.001** |
| **Age (years)** | | | | | | |
| •18–30 | 167 (87.4) | 24 (12.6) | 1 | | 1 | |
| •31–40 | 151 (83.0) | 31 (17.0) | 1.43 (0.80, 2.54) | 0.225 | 1.03 (0.53, 2.00) | 0.809 |
| •41–63 | 87 (71.9) | 34 (28.1) | 2.72 (1.52, 4.87) | **0.001** | 1.66 (0.83, 3.33) | 0.151 |
| **Education Level** | | | | | | |
| •No Education or Primary | 207 (88.5) | 27 (11.5) | 1 | | 1 | |
| •Secondary or above | 198 (76.2) | 62 (23.8) | 2.40 (1.47, 3.93) | **<0.001** | 1.67 (0.94, 2.97) | 0.079 |
| **Marital Status** | | | | | | |
| •Single | 76 (85.4) | 13 (14.6) | 1 | | 1 | |
| •Married | 277 (82.2) | 60 (17.8) | 1.27(0.66, 2.43) | 0.477 | 0.74 (0.34, 1.58) | 0.435 |
| •Divorced | 52 (76.5) | 16 (23.5) | 1.80 (0.80, 4.05) | 0.157 | 0.75 (0.28, 1.98) | 0.556 |
| **Country of Origin** | | | | | | |
| •Lao PDR | 146 (91.2) | 14 (8.8) | 1 | | | |
| •Cambodia | 119 (90.2) | 13 (9.8) | 1.14 (0.52, 2.52) | 0.747 | 1.72 (0.64, 4.66) | 0.285 |
| •Myanmar | 140 (69.3) | 62 (30.7) | 4.62 (2.47, 8.62) | **<0.001** | 4.68 (1.79, 12.21) | **0.002** |
| **Average Income (baht per month)** | | | | | | |
| •Less than or equal to 5000 | 67 (74.4) | 23 (25.6) | 1 | | 1 | |
| •5,001–10,000 | 138 (84.1) | 26 (15.9) | 0.55 (0.29, 1.03) | 0.063 | 1.12 (0.54, 2.30) | 0.754 |
| •10,001–15,000 | 166 (86.0) | 27 (14.0) | 0.47 (0.25, 0.88) | **0.019** | 1.43 (0.66, 3.10) | 0.369 |
| •15,001–30,000 | 34 (72.3) | 13 (27.7) | 1.11 (0.50, 2.47) | 0.791 | 1.53 (0.58, 3.99) | 0.389 |
| **Income or job loss during the COVID-19 pandemic** | | | | | | |
| •No | 190 (92.2) | 16 (7.8) | 1 | | 1 | |
| •Yes | 215 (74.7) | 73 (25.3) | 4.03 (2.27, 7.17) | **<0.001** | 3.99 (2.09, 7.62) | **<0.001** |
| **Ability to read in Thai Language** | | | | | | |
| •Cannot read at all | 185 (78.1) | 52 (21.9) | 1 | | 1 | |
| •Can read a little bit | 161 (88.0) | 22 (12.0) | 0.49 (0.28, 0.84) | **0.009** | 0.95 (0.49, 1.87) | 0.888 |
| •Can read fluently | 59 (79.7) | 15 (20.3) | 0.90 (0.47, 1.72) | 0.760 | 2.25 (0.86, 5.88) | 0.098 |
| **Ability to speak in Thai Language** | | | | | | |
| •Cannot speak at all | 31 (81.6) | 7 (18.4) | 1 | | – | |
| •Can speak a little bit | 162 (85.3) | 28 (14.7) | 0.77 (0.31, 1.91) | 0.566 | | |
| •Can speak fluently | 212 (79.7) | 54 (20.3) | 1.13 (0.47, 2.70) | 0.787 | | |
| **Migration status** | | | | | | |
| •Regular | 239 (81.0) | 56 (19.0) | 1 | | – | |
| •Irregular | 166 (83.4) | 33 (16.6) | 0.85 (0.53, 1.36) | 0.496 | | |

Notes. Model 1 = Simple Logistic Regression, Model 2 = Multiple Logistic Regression, COR = Crude odds ratio, AOR = Adjusted odds ratio, CI = Confidence Interval, α = 0.05. [a] p–value < 0.05 are in bold.

to women with a primary or lower education level (OR = 2.40, 95% CI = 1.47, 3.93). Compared to women who came from Lao PDR, those who came from Myanmar displayed a higher odds of reporting experiences of discrimination (OR = 4.62, 95% CI = 2.47, 8.62).

Women with an average monthly income of 10,001–15,000 baht showed lower odds of reporting being discriminated against than those who had a monthly income of 5,000 baht or less (OR = 0.47, 95% CI = 0.25, 0.88). Compared to women who did not lose income or their job during the pandemic, those who lost income or their job had higher odds of reporting experiences of discrimination (OR = 4.03, 95% CI = 2.27, 7.17). Women who were able to read the Thai language a little bit displayed lower odds of reporting being discriminated against than women who could not read it at all (OR = 0.49, 95% CI = 0.28, 0.84). The qualitative data confirmed that women migrant workers face language difficulties when they arrive in Thailand for the first time, struggling to make a living due to illiteracy. They hoped to study the Thai language to get more opportunities to find jobs. A lack of ability to use the Thai language also made migrant workers feel lonely, as they had limitations in communication.

"*Most of us couldn't speak Thai when I first arrived in Thailand. We were terrified. For me, I can't even afford to buy water. I went shopping and pointed out the items I needed...*" (FGD)

"*Language is an important issue. If you cannot speak Thai, it was very difficult for you to stay here...*" (In-depth interview: I-12)

"*I have been practicing for a year because I think if we can speak Thai, it's increasing our opportunity to find a job, I mean we can work anywhere if we can speak Thai. My mother cannot speak Thai, so she hardly understood while the supervisor asks her to do something, so she always be blamed, insulted. Mom always back home with her tears. Now I am teaching my mom to speak Thai, she asked me to teach speak Thai, so she speaks better...*" (In-depth interview: I-19)

After adjusting for potential predictor variables, multivariable analysis (Table 5) demonstrated that, compared to the respective counterparts, women who reported experiencing violence (AOR = 2.76, 95% CI = 1.49, 5.12), women who came from Myanmar (AOR = 4.68, 95% CI = 1.79, 12,21), and women who lost income or their job during the COVID-19 pandemic (AOR = 3.99, 95% CI = 2.09, 7.62) were more likely to report being discriminated against during the pandemic.

The qualitative data showed that most women participating in in-depth interviews and FGDs reported discrimination during the pandemic. Thematic analysis of the interviews uncovered several key themes related to experiences of discrimination among these women. First, women who experienced violence reported facing additional barriers and challenges in accessing support and resources during the pandemic. Many women shared that they were afraid to seek help due to concerns about their safety and privacy, which may have contributed to their increased risk of experiencing discrimination. Second, women from Myanmar reported experiencing discrimination based on their ethnicity and nationality. Participants shared that they faced negative attitudes and stereotypes from members of the broader community, as well as difficulty accessing healthcare and other essential services. Third, women who lost income or jobs during the pandemic reported feeling increasingly vulnerable and marginalized. Many women shared that they struggled to make ends meet and faced additional stressors and challenges, which may have contributed to their increased risk of discrimination.

## Challenge of women migrant workers impacted by the COVID-19 restrictions and coping mechanisms

During the COVID-19 pandemic, many women migrant workers faced profound challenges. Our study found that there were women, previously employed in the service industry who

endured abrupt job loss as businesses closed during lockdowns, plunging them into financial uncertainty. Some women migrants found themselves more susceptible to exploitation as job opportunities dwindled, highlighting the heightened risks faced by vulnerable populations during economic downturns. Many migrant women confronted obstacles in accessing essential services due to restrictions on movement and strained healthcare systems. The fear of contracting COVID-19 while seeking medical attention exacerbated the challenges she already faced. Below are the results from the qualitative part.

Disruptions in Employment: Women migrants shared their experience of sudden job loss due to the economic downturn caused by COVID-19. They had been working in the service industry, and the closure of businesses during lockdowns left them without a source of income. The uncertainty surrounding when they could return to work exacerbated financial stress and created challenges in meeting their basic needs.

Increased vulnerability to exploitation: Some women spoke about increased vulnerability to exploitation during the pandemic. With limited job opportunities, they found themselves in a precarious situation where employers took advantage of the high demand for work. The fear of losing their job made them hesitant to report mistreatment, highlighting the heightened risks faced by women migrant workers during times of economic uncertainty.

Challenges in accessing essential services: Restrictions on movement and overwhelmed healthcare systems made it difficult for women migrants to receive timely medical attention. The fear of contracting COVID-19 while seeking healthcare added an extra layer of stress to an already challenging situation, emphasizing the intersectionality of health, migration, and the pandemic.

In response to the challenges posed by the COVID-19 pandemic, this study uncovered a range of coping mechanisms employed by women migrant workers in Thailand. Quantitative data revealed that a significant number of respondents (137 women) reported actively seeking support from community networks, with 59.1% indicating reliance on peer assistance during times of distress. This quantitative insight was complemented by qualitative findings, this study emphasized the vital role of community solidarity in navigating uncertainties. Additionally, our qualitative findings highlighted individual resilience strategies, with many women engaging in skill-building activities during lockdowns to enhance their employability.

## Discussion

This study identified the extent of discrimination against women migrant workers from Cambodia, Lao PDR, and Myanmar in Thailand during the COVID-19 pandemic. Moreover, our study described the intersection of discrimination and violence and investigated the factors associated with discrimination against women migrant workers.

In this study, one in five women migrant workers (18.0%) self-identified that they were discriminated against during the COVID-19 pandemic; discrimination for migrant worker status was found to be the most common form. Comparatively, a study on Thai public attitudes towards migrant workers found that 53% believed the country did not need low-skilled migrants, 40% saw them as an economic burden, and 38% thought they negatively impact the overall economy [16]. During the pandemic, migrant workers were often "stigmatized and stereotyped" by native individuals, and some employers announced that they did not accept migrant workers. They faced "nationality-based discrimination" because of exclusive enforcement that restricted their travel from one province to another and "racial discrimination" by authorities or officials when accessing health services [24]. Because the assistance provided by the Thai government during the pandemic targeted mainly their citizens, and because laws and policies were based on the interests of national security, migrant workers were neglected

and faced problems related to legal status, such as expiring work permits or no longer having an employer [25].

In addition, our study revealed that women migrant workers in Thailand encountered several forms of violence, most commonly psychological and economic violence during the COVID-19 pandemic. The pandemic situation in Thailand fostered a perception that migrant workers were contributing negatively to the spread of the virus. Additionally, the economic downturn caused by the pandemic led to job losses and challenges in the legal status of migration for these workers. Consequently, their vulnerability increased, and their access to health and legal support became limited [3–5].

This study revealed the interesting finding that two-thirds of women migrant workers who reported experiencing discrimination had also experienced intimate partner violence (IPV). Additionally, three-fourths of women who reported discrimination were survivors of non-intimate partner violence (N-IPV) at some point in their lives. This finding highlights the intersectionality of discrimination and violence among women migrant workers as suggested by intersectionality theory that provides a framework for understanding how various forms of discrimination, along with violence and other social characteristics such as country of origin and occupation, intersect and influence the experiences of women migrant workers due to the overlap of these identities [12]. In addition, our finding aligns with social learning theory which provides a concept that women migrant workers might also learn to expect negative behavior particularly discrimination from their social surroundings if they have already experienced violence in their lives [13]. It suggests that discrimination and violence are not isolated experiences but rather intersecting factors that contribute to the vulnerability of migrant women workers [26].

Due to travel restrictions and border closures during the pandemic, women migrant workers faced the intersecting challenges of economic difficulties and reduced job opportunities. These challenges forced them to use irregular migration channels and risky smugglers, which made them more vulnerable to engaging in unsafe employment [27]. The pandemic also exacerbated their vulnerability to violence and discrimination in the workplace. The lockdown measures further isolated them and reduced their capacity to escape unfavorable employment situations [28]. These circumstances created a perfect storm for migrant women workers, who faced multiple layers of vulnerability and marginalization during the pandemic.

As hypothesized, this study found that women migrant workers who reported experiencing violence in their lifetime were more likely to have experienced discrimination during the pandemic. This result is supported by findings from previous studies that identified an association between discrimination and violence against women [29–31]. This may be due to the intersection of discrimination and violence [12]. Other possible reasons for this association are that women who face violence in their life may be tolerant of discrimination [13] or that women who are survivors of violence may be perceived as being less efficient and less entitled to respect [14]. Another possible explanation for the higher likelihood of discrimination among women who have experienced violence is that they may have been more vulnerable and faced additional barriers to accessing resources and support during the pandemic, which may have led to further marginalization and vulnerability during the pandemic.

The bivariate and multivariable models showed that women migrant workers from Myanmar were more likely to be subjected to discrimination. One previous study of Thai public opinion on migrants from Myanmar found that Thai people were deeply concerned with security issues and the possibility that migrant workers from Myanmar might endanger public safety or carry diseases. In that study, Thai people also believed that migrants from Myanmar were competing with native Thais for jobs and the country's resources [32]. Several factors, including historical tensions between Myanmar and Thailand and negative stereotypes and

biases toward Burmese people, may have contributed to this finding. This finding highlights that intervention strategies for the prevention of and response to discrimination in the context of migrant workers may be tailored based on their country of origin.

Logistic regression analyses identified that women migrant workers who lost income or jobs during the pandemic were more likely to experience discrimination. Women migrant workers were more vulnerable to the COVID-19 pandemic compared to men in terms of economic and social aspects. Several challenges of the pandemic situation, such as decreased income and higher expenses, increased their vulnerability [24]. Women who lost their jobs or had their income reduced may have been more likely to experience discrimination as they struggled to make ends meet and faced additional stressors and challenges during the pandemic. When women migrant workers became economically weakened because of job or income loss, it may have worsened power imbalances with their partners, families, and communities, leading to more discrimination against them. They may have been discriminated against when seeking other employment or requesting financial assistance, being perceived as less valuable or less skilled.

In accordance with the findings of previous studies [33,34], this study showed that women migrant workers of older age showed a higher likelihood of being discriminated against. Ageism against older women migrant workers may be due to perceptions of their low productivity, which may lead to further discrimination in an intersection with racism or sexism. In this study, women with a higher education level displayed higher odds of being discriminated against. This finding contradicts the findings of a previous study [35], which revealed that a higher educational level is associated with lower discrimination, but it is aligned with the findings of other studies [36,37], which identified that discrimination increased with educational level. Different contextual backgrounds and different study populations may have contributed to these inconsistent findings. Bivariate analysis showed that women who were able to read Thai had lower odds of experiencing discrimination. This was supported by the qualitative finding that migrant women were more vulnerable to discrimination if their Thai language skills were limited or absent, as this reduced their ability to report on violence committed against them in the host society.

This study is the first mixed-methods study conducted in Thailand about discrimination against women migrant workers from Cambodia, Lao PDR, and Myanmar during the COVID-19 pandemic. The mixed-methods design could provide a better understanding of the intersection of discrimination and violence. However, the respondents' perceived experience of discrimination was asked about using a "yes" or "no" question. Some participants might not have been aware of discrimination against them, as the scope of this study was limited to three aspects, which might have omitted other forms of discrimination. The result might not be generalizable to the larger population of women migrant workers in Thailand because data were collected in selected provinces. The snowball sampling technique might have led to selection bias, and the study sample might not be representative of the whole study population. Women with experiences of violence and discrimination might have been more likely to participate in the study to share their experiences. Interviews conducted by enumerators might have limited the explanation of questions and data collection. Self-reporting bias and recall bias might also have occurred. However, these limitations were restricted by proper training and close monitoring of the enumerators by the researchers, by maintaining the privacy of and rapport with participants during the data collection, and by collaboration with local NGOs and CSOs.

## Recommendations and conclusion

As this study found that a significant proportion of women migrant workers had irregular migration status, laws and policies about the process of recruiting migrant workers should be

oriented toward safer migration, and accessible channels should be ensured for the provision of documentation status. Cross-border linkages between the authorities of countries should be strengthened to support safer migration through coordination and cooperation at different levels of stakeholders. Moreover, to address the intersecting forms of discrimination and violence, it is recommended that policymakers and service providers provide services that are inclusive, responsive, and gender-sensitive to the unique situations of women migrants workers to ensure their accessibility and availability. Women migrant workers who are survivors of violence, those with low incomes, and those coming from Myanmar should be prioritized because they were found to have a higher risk of facing discrimination. In addition, peer networks, awareness-raising education, and training programs should be strengthened and adapted by applying feedback from women migrant workers. Ultimately, it is recommended that future studies be conducted on a regular basis for the monitoring and analysis of up-to-date situations. Overall, the results of this study highlight the need for greater attention to the intersection of discrimination and violence against women, particularly among migrant workers who may face multiple forms of vulnerability and marginalization. Addressing discrimination and violence against women will require a multifaceted approach that addresses structural factors, such as gender inequality, as well as individual- and community-level interventions aimed at promoting gender equality, preventing violence, and supporting survivors.

## Supporting information

**S1 File. Interview survey questionnaire of this study.**
(PDF)

**S2 File. Quantitative dataset of this study.**
(XLSX)

**S3 File. STROBE checklist of this study.**
(DOCX)

## Author Contributions

**Conceptualization:** Montakarn Chuemchit, Ratana Somrongthong.

**Data curation:** Montakarn Chuemchit, Nyan Linn, Suttharuethai Chernkwanma.

**Formal analysis:** Montakarn Chuemchit, Nyan Linn, Chit Pyae Pyae Han, Zayar Lynn, Nutta Taneepanichskul, Wandee Sirichokchatchawan.

**Funding acquisition:** Montakarn Chuemchit.

**Investigation:** Montakarn Chuemchit, Suttharuethai Chernkwanma.

**Methodology:** Montakarn Chuemchit, Nyan Linn, Suttharuethai Chernkwanma, Nutta Taneepanichskul, Wandee Sirichokchatchawan.

**Project administration:** Suttharuethai Chernkwanma.

**Supervision:** Montakarn Chuemchit, Ratana Somrongthong.

**Validation:** Montakarn Chuemchit.

**Visualization:** Montakarn Chuemchit, Nyan Linn, Chit Pyae Pyae Han, Zayar Lynn, Nutta Taneepanichskul, Wandee Sirichokchatchawan.

**Writing – original draft:** Montakarn Chuemchit, Nyan Linn.

**Writing – review & editing:** Montakarn Chuemchit, Nyan Linn.

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
