## [Decision Letter · Decision Letter 0]

25 Jan 2024

PONE-D-23-19289Discrimination and violence against women migrant workers in Thailand during COVID-19: A mixed methods studyPLOS ONE

Dear Dr. Chuemchit,

Thank you for submitting your manuscript to PLOS ONE. After careful consideration, we feel that it has merit but does not fully meet PLOS ONE’s publication criteria as it currently stands. Therefore, we invite you to submit a revised version of the manuscript that addresses the points raised during the review process.

We look forward to receiving your revised manuscript.

Kind regards,

Kyaw Lwin Show, MPH, PhD

Academic Editor

PLOS ONE

Journal Requirements:

3. Please provide additional information regarding the considerations made for the migrants included in this study. For instance, please discuss whether participants were able to opt out of the study and whether individuals who did not participate receive the same treatment offered to participants.

Additional Editor Comments:

Generally, it is a well written paper. Please

- provide an explanation for the choice of outcome variable 'discrimination' in Table 5 instead of 'violence'.

- response the reviewers' comments.

Reviewers' comments:

Reviewer's Responses to Questions

**Comments to the Author**

1. Is the manuscript technically sound, and do the data support the conclusions?

Reviewer #1: Yes

Reviewer #2: Yes

2. Has the statistical analysis been performed appropriately and rigorously? 

Reviewer #1: Yes

Reviewer #2: No

3. Have the authors made all data underlying the findings in their manuscript fully available?

Reviewer #1: Yes

Reviewer #2: No

4. Is the manuscript presented in an intelligible fashion and written in standard English?

Reviewer #1: Yes

Reviewer #2: Yes

5. Review Comments to the Author

Reviewer #1: This paper describes the important issue of discrimination and migration issue interconnected with violence. However, the main variable of COVID 19 compounding factor was not explicitly seen and it is hard to say is because of COVID 19 and restrictions. If the qualitative findings would be added more for the COVID 19 restrictions and impact, the paper would be better. Although the authors have added the limitations, we must be careful in drawing conclusions of the interconnectedness of discrimination, violence and COVID 19 period. Main issue facing migrant populations are unique and subjective. I also did not see any findings on coping mechanism from both quantitative and qualitative findings.

Line 32 violence (Is it both IPV and NIPV)

Line 226 it should be four in ten but I think ten is missing in the sentence.

Discrimination is self-identified as 18 percent and in the discussion how is it different from other studies.

Reviewer #2: This research work highlights the importance of mitigating the risks and challenges being encountered by women migrant workers in the time of public health emergency and has policy and program implications by elucidating the nexus of discrimination and violence. Please do minor but essential revisions to improve scientific integrity.

Title:

- To revise as : "Discrimination and violence against women migrant workers in Thailand during

the COVID-19 pandemic: A mixed-methods study"

1. Introduction

- To shorten the introduction section by deleting the first two paragraphs [LINES 45-60].

- To specify whether the investigators were hypothesizing : "women migrant workers who have had experiences of

violence are more likely to face discrimination in the context of COVID-19 pandemic" [LINES 82-83]

- To add "Intersectionality theory uncovers the sociological analytical framework that explains-----------the challenges and

experiences of women migrant workers". [LINES 83-85].

- To add whether there is an emergency regulatory measures during this public health crisis in Thailand that there will be

no legal action towards women migrant workers with irregular migration status if they are approaching for screening,

testing and treatment measures to mitigate the spread of virus.

2. Data collection by enumerators

- To add the number and qualifications of enumerators per province and the duration of training arranged.

3. Research instrument

- To add whether there was a pilot testing for the qualitative research instruments or not.

4. Data analysis

- To clarify Model 1: Is it similar to cross tabulations (bivariate analysis) resulting in crude odds ratio and 95% CI? In my

opinion, it is not a simple logistic regression model.

5. Results

- To add the median age and its interquartile range in Table 2 instead of mean and SD.

- To add any relevant qualitative excerpt (s) to understand Table 3 in addition to the summary paragraph of qualitative

data.

- To compute KAPPA statistic and significance level to explain the Intersection of discrimination and experiences of violence

of women migrant workers (Table 4).

- To add one table depicting the comprehensive information generated from the structured questionnaire regarding the

five parts for intimate and non intimate partner violence particularly within past 12 months coinciding with COVID-19

pandemic. Further interpretations and the relevant discussion is mandatory.

- To clarify the reason of not using the number of women who reported any form of violence within past 12 months in

bivariate and multivariate analyses.

6. Discussion

- To specify how the Intersectionality theory and Social Learning Theory support the data, interpretations and conclusion.

6. PLOS authors have the option to publish the peer review history of their article (what does this mean?). If published, this will include your full peer review and any attached files.

Reviewer #1: No

Reviewer #2: **Yes: **KHIN-THET-WAI

---

## [Author Response · Author response to Decision Letter 0]

16 Feb 2024

Response of authors to comments from editors and reviewers on the manuscript “Discrimination and violence against women migrant workers in Thailand during the COVID-19 pandemic: A mixed-methods study”

Note: the line number mentioned in this response sheet are intended to refer to “the manuscript with tract changes.”

Thank you very much for the comment. The authors made sure that this manuscript meets PLOS ONE’s requirements.

- Please provide additional information regarding the considerations made for the migrants included in this study. For instance, please discuss whether participants were able to opt out of the study and whether individuals who did not participate receive the same treatment offered to participants.

Thank you for this important comment. The authors wrote in Line 160 – “At any time during the interview, the respondents were able to pause, skip the question, or stop answering” and in Line 215 – “Participants were kept anonymous, and their involvement was completely voluntary”. For better clarification, we added additional information regarding the considerations made for the migrants included in this study in the manuscript Line 217 – 220 as follows “Participants had the option to withdraw from the study at any point during the recruitment and interview process. Furthermore, those individuals who were recruited but did not complete the interview received the same treatment offered to respondents who successfully participated in the study.” To address your concern about the treatment of individuals who did not participate, we want to emphasize that we are committed to ensuring equitable consideration for all individuals involved in the study. In addition to the information provided in Lines 217 – 220, we would like to highlight that individuals who were recruited but did not complete the interview were still offered the same level of support and access to resources as the participants who successfully took part in the study. As part of our commitment to ethical research practices and the well-being of all individuals involved, we have compiled a comprehensive list of helping services available in both the local communities and Thailand. This includes resources that can provide support, guidance, and assistance tailored to the needs of migrants. This list is intended to serve as a valuable resource for those who may choose not to participate or complete the interview but still require assistance. We want to assure you that our approach extends beyond the study itself to support the overall well-being of the individuals involved.

- Please include captions for your Supporting Information files at the end of your manuscript, and update any in-text citations to match accordingly. 

Thank you for this comment. This manuscript does not include supporting information files, as all pertinent information is thoroughly presented within the main text. We also added “Supporting information” at the end of the manuscript at Line 628 – 630 that “This manuscript does not include supporting information files, as all pertinent information is thoroughly presented within the main text.”

Additional Editor Comments:

- Generally, it is a well written paper. Please provide an explanation for the choice of outcome variable 'discrimination' in Table 5 instead of 'violence'.

Thank you for your comment. We mentioned in the manuscript in Line 83 – 84 that “This study hypothesizes that women migrant workers who have had experiences of violence are more likely to face discrimination” and the objectives of this paper are written in Line 115 – 118 that “to investigate 1) the extent of discrimination against women migrant workers in Thailand during the pandemic, 2) the intersection of this discrimination with these women’s experiences of violence, and 3) the factors associated with discrimination.” Specifically in this paper, we primarily focus on the outcome “discrimination,” in accordance with the research objectives. Therefore, to achieve the objective 3 of this study, we chose the outcome variable “discrimination” in Table 5 to find its associated factors. 

- Response the reviewers' comments.

Thank you for the comment. The author responded to the comments from reviewers.

Comments to the Author

1. Is the manuscript technically sound, and do the data support the conclusions?

- Reviewer #1: Yes

- Reviewer #2: Yes

2. Has the statistical analysis been performed appropriately and rigorously?

- Reviewer #1: Yes

- Reviewer #2: No

3. Have the authors made all data underlying the findings in their manuscript fully available? In addition to summary statistics, the data points behind means, medians and variance measures should be available. If there are restrictions on publicly sharing data—e.g. participant privacy or use of data from a third party—those must be specified.

- Reviewer #1: Yes

- Reviewer #2: No

Thank you for your comment. We added “Data availability statement” in the manuscript in Line 632 – 634 “The datasets used and analyzed during the current study are available from the corresponding author upon reasonable request.”

Comments from reviewers

Reviewer #1: 

1. This paper describes the important issue of discrimination and migration issue interconnected with violence. However, the main variable of COVID 19 compounding factor was not explicitly seen and it is hard to say is because of COVID 19 and restrictions. If the qualitative findings would be added more for the COVID 19 restrictions and impact, the paper would be better. Although the authors have added the limitations, we must be careful in drawing conclusions of the interconnectedness of discrimination, violence and COVID 19 period. Main issue facing migrant populations are unique and subjective. I also did not see any findings on coping mechanism from both quantitative and qualitative findings.

 Thank you for the comment. The authors would like to express gratitude to the reviewer. We revised the main text in Line 368 – 393 regarding “Challenge of women migrant workers impacted by the COVID-19 restrictions and coping mechanisms” as follows.

“During the COVID-19 pandemic, many women migrant workers faced profound challenges. Our study found that there were women, previously employed in the service industry who endured abrupt job loss as businesses closed during lockdowns, plunging them into financial uncertainty. Some women migrants found themselves more susceptible to exploitation as job opportunities dwindled, highlighting the heightened risks faced by vulnerable populations during economic downturns. Many migrant women confronted obstacles in accessing essential services due to restrictions on movement and strained healthcare systems. The fear of contracting COVID-19 while seeking medical attention exacerbated the challenges she already faced. Below are the results from the qualitative part. 

Disruptions in Employment: Women migrants shared their experience of sudden job loss due to the economic downturn caused by COVID-19. They had been working in the service industry, and the closure of businesses during lockdowns left them without a source of income. The uncertainty surrounding when they could return to work exacerbated financial stress and created challenges in meeting their basic needs.

Increased Vulnerability to Exploitation: Some women spoke about increased vulnerability to exploitation during the pandemic. With limited job opportunities, they found themselves in a precarious situation where employers took advantage of the high demand for work. The fear of losing their job made them hesitant to report mistreatment, highlighting the heightened risks faced by women migrant workers during times of economic uncertainty.

Challenges in Accessing Essential Services: Restrictions on movement and overwhelmed healthcare systems made it difficult for women migrants to receive timely medical attention. The fear of contracting COVID-19 while seeking healthcare added an extra layer of stress to an already challenging situation, emphasizing the intersectionality of health, migration, and the pandemic”. 

Regarding the Coping mechanisms, we added in the manuscript Line 394 – 401 “In response to the challenges posed by the COVID-19 pandemic, this study uncovered a range of coping mechanisms employed by women migrant workers in Thailand. Quantitative data revealed that a significant number of respondents (137; 50.9% of survivors) reported actively seeking support from community networks, with 59.1% indicating reliance on peer assistance during times of distress. This quantitative insight was complemented by qualitative findings, this study emphasized the vital role of community solidarity in navigating uncertainties. Additionally, our qualitative findings highlighted individual resilience strategies, with many women engaging in skill-building activities during lockdowns to enhance their employability”.

2. Line 32 violence (Is it both IPV and NIPV)

Thank you for the question. In Line 32, “violence” refers to any violence (either IPV or NIPV or both). For better clarification, we revised it in Line 33 as “The multivariable analysis revealed that women migrant workers who had experienced any violence (AOR = 2.76, 95% CI = 1.49, 5.12).”

3. Line 226 it should be four in ten but I think ten is missing in the sentence.

Thank you for raising this point. The authors are sorry for this typing error, and we revised it in Line 246 as “and four in 10 in women had irregular migration status…”.

4. Discrimination is self-identified as 18 percent and in the discussion how is it different from other studies. 

Thank you for the comment. The authors revised by adding the discussion in Line 407 “In this study, one in five women migrant workers (18.0%) self-identified that they were discriminated against during the COVID-19 pandemic.” We compared this finding with previous studies in Line 409 – 412 “Comparatively, a study on Thai public attitudes towards migrant workers found that 53% believed the country did not need low-skilled migrants, 40% saw them as an economic burden, and 38% thought they negatively impact the overall economy.”

Reviewer #2: 

1. This research work highlights the importance of mitigating the risks and challenges being encountered by women migrant workers in the time of public health emergency and has policy and program implications by elucidating the nexus of discrimination and violence. Please do minor but essential revisions to improve scientific integrity.

Thank you for your supportive and encouraging comment. The authors would like to acknowledge to the reviewer. We did the revisions according to the comments.

2. Title: To revise as : "Discrimination and violence against women migrant workers in Thailand during the COVID-19 pandemic: A mixed-methods study"

Thank you for the comment. We revised the title in Line 1-2 “Discrimination and violence against women migrant workers in Thailand during the COVID-19 pandemic: A mixed-methods study”.

1. Introduction

3. To shorten the introduction section by deleting the first two paragraphs [LINES 45-60].

Thank you for the comment. We revised the introduction to shorten it by deleting the first two paragraphs in Line 46 – 61.

4. To specify whether the investigators were hypothesizing : "women migrant workers who have had experiences of violence are more likely to face discrimination in the context of COVID-19 pandemic" [LINES 82-83]

Thank you so much for this comment. We revised it according to the suggestion in Line 83 – 84 “This study hypothesizes that women migrant workers who have had experiences of violence are more likely to face discrimination in the context of COVID-19 pandemic.”

5. To add "Intersectionality theory uncovers the sociological analytical framework that explains-----------the challenges and experiences of women migrant workers". [LINES 83-85].

We greatly appreciate this comment. We revised it according to the suggestion in Line 85 “Intersectionality theory uncovers the sociological analytical framework that explains…”

6. To add whether there is an emergency regulatory measures during this public health crisis in Thailand that there will be no legal action towards women migrant workers with irregular migration status if they are approaching for screening, testing and treatment measures to mitigate the spread of virus.

Thank you for the comment. We revised by adding in the manuscript in Line 103 – 109 “In the context of the COVID-19 crisis in Thailand, it is important to clarify that there were no emergency regulatory measures in place that would result in legal action against women migrant workers with irregular migration status seeking screening, testing, or treatment measures to mitigate the spread of the virus. While they were encouraged to approach for assistance without fear of immediate legal consequences related to irregular migration, it is essential to note that the issue of irregular migration status remains a separate matter.”

2. Data collection by enumerators

7. To add the number and qualifications of enumerators per province and the duration of training arranged.

Thank you for the comment. The authors added in the manuscript Line 143 – 144 “There were 5-8 enumerators per province depending on the number of respondents.” And in Line 144 – 145 “The team conducted in-person and virtual 2-day workshop training…”. And in Line 156 – 160 “Moreover, during data collection, the researchers and enumerators held wrap-up meetings to discuss the day's data collection activities. These meetings provided an opportunity to review and discuss the data collection process, address any issues that arose, and ensure consistency and accuracy in the overall data collection effort.”

3. Research instrument

8. To add whether there was a pilot testing for the qualitative research instruments or not.

Thank you for the comment. We revised the manuscript in Line 206 – 209 “We conducted pilot testing to evaluate and refine the qualitative research instruments and methodologies prior to the full-scale investigation. During this preliminary stage, a small group of representative participants was recruited to participate in interviews, allowing for an assessment of the instruments’ clarity, cultural sensitivity, and effectiveness.”

4. Data analysis

9. To clarify Model 1: Is it similar to cross tabulations (bivariate analysis) resulting in crude odds ratio and 95% CI? In my opinion, it is not a simple logistic regression model.

Thank you for the question. We did cross-tabulation in Stata using the command “tab ind_var dep_var” to get the cross-tabulation between independent variables and discrimination in the first two columns of Table 7. However, if cross-tabulation is used together with chi-square in bivariate analysis, it could not result in a crude odds ratio and 95% CI because some of the independent variables had more than 3 categories (e.g., age, marital status). Therefore, instead of using the chi-square test, we analyzed simple logistic regression using the command “logistic dep_var ind_var” in Stata for bivariate analysis in model 1 resulting in crude odds ratios and 95% CI. For better clarification, we revised the manuscript in Line 229 and Table 7. 

5. Results

10. To add the median age and its interquartile range in Table 2 instead of mean and SD.

Thank you for the comment. We revised by adding the median age and its interquartile range in Table 2 instead of the mean and SD in Line 224 – 225, 242.

11. To add any relevant qualitative excerpt (s) to understand Table 3 in addition to the summary paragraph of qualitative data.

We appreciate this comment. The manuscript was revised by adding in Line 269 – 275 “These findings are consistent with the qualitative results of the study regarding the respondent's experiences of discrimination, encompassing aspects related to their migrant worker status, gender, and age. The narratives provided by participants reveal multifaceted encounters with discrimination that shed light on the intersectionality of these factors. For instance, several respondents expressed facing discrimination not only due to their migrant status but also as a consequ

---

## [Decision Letter · Decision Letter 1]

27 Feb 2024

Discrimination and violence against women migrant workers in Thailand during COVID-19: A mixed methods study

PONE-D-23-19289R1

Dear Dr. Chuemchit,

We’re pleased to inform you that your manuscript has been judged scientifically suitable for publication and will be formally accepted for publication once it meets all outstanding technical requirements.

Kind regards,

Kyaw Lwin Show, MPH, PhD

Academic Editor

PLOS ONE

Additional Editor Comments (optional):

Reviewers' comments:

Reviewer's Responses to Questions

**Comments to the Author**

1. If the authors have adequately addressed your comments raised in a previous round of review and you feel that this manuscript is now acceptable for publication, you may indicate that here to bypass the “Comments to the Author” section, enter your conflict of interest statement in the “Confidential to Editor” section, and submit your "Accept" recommendation.

Reviewer #2: All comments have been addressed

2. Is the manuscript technically sound, and do the data support the conclusions?

Reviewer #2: Yes

3. Has the statistical analysis been performed appropriately and rigorously? 

Reviewer #2: Yes

4. Have the authors made all data underlying the findings in their manuscript fully available?

Reviewer #2: Yes

5. Is the manuscript presented in an intelligible fashion and written in standard English?

Reviewer #2: Yes

6. Review Comments to the Author

Reviewer #2: This research work highlights the importance of mitigating the risks and challenges being encountered by women migrant workers in the time of public health emergency and has policy and program implications by elucidating the nexus of discrimination and violence.

There are no additional comments for the revised version.

7. PLOS authors have the option to publish the peer review history of their article (what does this mean?). If published, this will include your full peer review and any attached files.

Reviewer #2: **Yes: **KHIN-THET-WAI
